# *PointNSP*: Autoregressive 3D Point Cloud Generation with Next-Scale Level-of-Detail Prediction

## Abstract

Autoregressive point cloud generation has long lagged behind diffusion-based approaches in quality. The performance gap stems from the fact that autoregressive models impose an artificial ordering on inherently unordered point sets, forcing shape generation to proceed as a sequence of local predictions. This sequential bias emphasizes short-range continuity but undermines the model's capacity to capture long-range dependencies, hindering its ability to enforce global structural properties such as symmetry, consistent topology, and large-scale geometric regularities. Inspired by the level-of-detail (LOD) principle in shape modeling, we propose *PointNSP*, a coarse-to-fine generative framework that preserves global shape structure at low resolutions and progressively refines fine-grained geometry at higher scales through a next-scale prediction paradigm. This multi-scale factorization aligns the autoregressive objective with the permutation-invariant nature of point sets, enabling rich intra-scale interactions while avoiding brittle fixed orderings. Experiments on ShapeNet show that *PointNSP* establishes state-of-the-art (SOTA) generation quality for the first time within the autoregressive paradigm. In addition, it surpasses strong diffusion-based baselines in parameter, training, and inference efficiency. Finally, in dense generation with 8,192 points, *PointNSP*'s advantages become even more pronounced, underscoring its scalability potential.

## 1 Introduction

Point clouds are a fundamental representation of 3D object shapes, describing each object as a collection of points in Euclidean space. They arise naturally from sensors such as LiDAR and laser scanners, offering a compact yet expressive encoding of fine-grained geometric details. Developing powerful generative models for point clouds is key to uncovering the underlying distribution of 3D shapes, with broad applications in shape synthesis, reconstruction, computer-aided design, and perception for robotics and autonomous systems. However, high-fidelity point cloud generation remains challenging due to the irregular and unordered nature of point

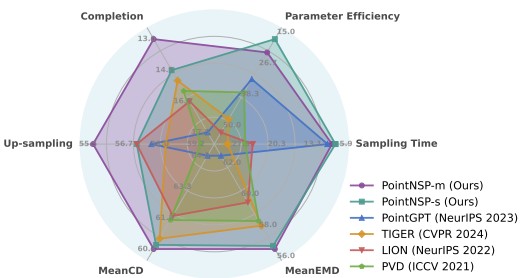

Figure 1: A comprehensive performance comparison between *PointNSP* and recent strong baseline methods across six key metrics.

sets (Qi et al., 2017a;b; Zaheer et al., 2017). Unlike images or sequences, point clouds have no inherent ordering—permuting the points leaves the shape unchanged—making naive order-dependent methods poorly suited for their generation and analysis.

In recent years, diffusion-based methods (Zhou et al., 2021; Zeng et al., 2022; Ren et al., 2024) have become the dominant paradigm for 3D point cloud generation, delivering strong results. However, their Markovian assumption overlooks global context, often leading to incoherent shapes. Moreover, diffusion models are computationally costly, as generating high-quality samples typically requires hundreds to thousands of denoising steps—a burden that grows prohibitive when producing dense point clouds. By contrast, autoregressive (AR) models condition on the full history, enabling explicit

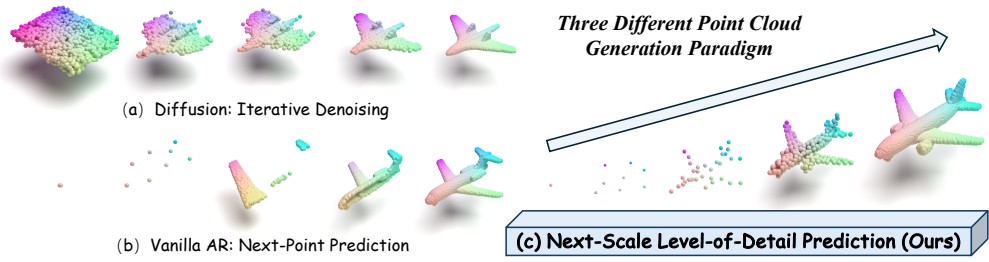

Figure 2: Three types of point cloud generative models: (a) diffusion-based methods that iteratively denoise shapes starting from Gaussian noise; (b) vanilla autoregressive (AR) methods that predict the next point by flattening the 3D shape into a sequence; and (c) our proposed *PointNSP*, which predicts next-scale level-of-detail in a coarse-to-fine manner.

modeling of long-range dependencies while generally offering faster sampling. Unlike diffusion models, existing AR methods require flattening inherently unordered point clouds into sequences. Point Transformers (Zhao et al., 2021; Wu et al., 2022; 2024) design specialized architectures for unordered point sets and explore diverse point cloud serialization strategies to improve speed, at the expense of relaxing permutation invariance. PointGrow (Sun et al., 2020) enforces a sequential order by sorting points along the $z$-axis. ShapeFormer (Yan et al., 2022) voxelizes point clouds and flattens codebook embeddings into sequences using a row-major order. PointVQVAE (Cheng et al., 2022) projects patches onto a sphere and arranges them in a spiral sequence to establish a canonical mapping. AutoSDF (Mittal et al., 2022) treats point clouds as randomly permuted sequences of latent variables, while PointGPT (Chen et al., 2023) leverages Morton code ordering to impose structure on unordered data. Although these approaches yield promising results, they still lag behind state-of-the-art diffusion-based methods (Luo & Hu, 2021; Zhou et al., 2021; Zeng et al., 2022; Ren et al., 2024) in generation quality. This is largely due to the restrictive unidirectional dependencies imposed by fixed sequential orders, which collapse global shape generation into local predictions and inherently violate the fundamental permutation-invariance property. This naturally raises the question: *can we achieve permutation-invariant autoregressive modeling for 3D point cloud generation?*

In this work, we introduce *PointNSP*, a novel autoregressive framework for 3D point cloud generation that preserves global permutation invariance—a key property ensuring that shapes remain independent of point ordering. Inspired by visual autoregressive modeling in 2D image synthesis (Tian et al., 2024), *PointNSP* follows a coarse-to-fine strategy, progressively refining point clouds from global structures to fine-grained details via next-scale prediction. Unlike prior approaches that predict one point at a time (next-point prediction), *PointNSP* captures multiple levels of detail (LoD) (Luebke et al., 2003) at each step, enabling more effective modeling of both global geometry and local structure. This design offers two key advantages. First, it avoids collapsing 3D structures by eliminating the need to flatten point clouds into 1D sequences: each step corresponds to a full 3D shape at a given LoD, ensuring structural coherence and permutation invariance. Second, compared to diffusion-based methods, *PointNSP* establishes a more structured and efficient generation trajectory, avoiding iterative noise injection and denoising in 3D space. Together, these advances allow *PointNSP* to achieve high generation quality while maintaining strong modeling efficiency. The comparisons across different paradigms are illustrated in Figure 2.

We conduct extensive experiments on the ShapeNet benchmark to validate the effectiveness of *PointNSP* across diverse settings. In the standard single-class scenario, *PointNSP* achieves state-of-the-art (SOTA) **generation quality**, yielding the lowest average Chamfer Distance and Earth Mover's Distance—setting a new benchmark for autoregressive modeling. Beyond quality, *PointNSP* also demonstrates substantially higher **parameter efficiency**, **training efficiency**, and **sampling speed** compared to strong diffusion-based baselines. In the more challenging multi-class setting, *PointNSP* maintains SOTA performance, evidencing superior cross-category generalization. Moreover, *PointNSP* significantly outperforms existing approaches on downstream tasks such as partial point cloud completion and upsampling, further highlighting the robustness and versatility of its design. Comparative results across these metrics are presented in Figure 1. When evaluated on denser point clouds with 8192 points, the advantages of *PointNSP* become even more pronounced, particularly in the aforementioned efficiency metrics, underscoring its **scalability potential**.

## 2 RELATED WORKS

**Autoregressive Generative Modeling.** The core principle of autoregressive generative models is to synthesize outputs sequentially by iteratively generating intermediate segments. This paradigm has achieved remarkable success in discrete language modeling through next-token prediction (Patel et al., 2023; OpenAI, 2023). Inspired by these advances, researchers have extended autoregressive modeling to other modalities, including images (Esser et al., 2021; Lee et al., 2022; Dosovitskiy et al., 2021), speech (Zhang et al., 2023; Wu et al., 2023a), and multi-modal data (Zhou et al., 2024; Team, 2024). Although these modalities are often continuous in nature, they are typically transformed into discrete token representations using techniques such as VQ-VAE (VQ) (van den Oord et al., 2017) or residual vector quantization (RVQ) (Lee et al., 2022), with generation performed over the resulting tokens in predefined orders (e.g., raster-scan sequences). To relax the constraint of strict unidirectional dependencies, MaskGIT (Chang et al., 2022) predicts sets of randomly masked tokens at each step under the control of a scheduler. More recently, VAR (Tian et al., 2024) redefines the autoregressive paradigm by predicting the next resolution rather than the next token. Within each scale, a bidirectional transformer enables full contextual interaction among tokens, making VAR especially effective for modeling unordered data such as 3D point clouds.

**Point Cloud Generation.** Deep generative models have made significant strides in 3D point cloud generation. For instance, PointFlow (Yang et al., 2019) captures the latent distribution of point clouds using continuous normalizing flows (Rezende & Mohamed, 2015; Grathwohl et al., 2019; Chen et al., 2018). Building on this foundation, a series of approaches—including DPM (Luo & Hu, 2021), ShapeGF (Cai et al., 2020), PVD (Zhou et al., 2021), LION (Zeng et al., 2022), TIGER (Ren et al., 2024), PDT (Wang et al., 2025)—leverage denoising diffusion probabilistic models (Ho et al., 2020; Song et al., 2021b) to synthesize 3D point clouds through the gradual denoising of input data or latent representations in continuous space. While efforts to improve the sampling efficiency of diffusion models—such as straight flows (Lipman et al., 2023; Liu et al., 2023; Wu et al., 2023b) and ODE-based solvers (Song et al., 2021a)—have achieved acceleration, these methods often introduce trade-offs that compromise generation quality. In contrast, autoregressive models for 3D point cloud generation (Sun et al., 2020; Cheng et al., 2022; Chen et al., 2023) have received relatively less attention, historically lagging behind diffusion-based techniques in terms of fidelity. In this work, we reformulate point cloud generation as an iterative upsampling process, establishing a connection with sparse point cloud upsampling approaches (Yu et al., 2018; He et al., 2023; Qu et al., 2024). A more comprehensive review of related point cloud upsampling methods is provided in the Appendix F.

## 3 METHOD: POINTNSP

### 3.1 PRELIMINARIES: AUTOREGRESSIVE 3D POINT CLOUD GENERATION

A point cloud is represented as a set of $N$ points $\mathbf{X} = \{\mathbf{x}_i\}_{i=1}^N$, where each point $\mathbf{x}_i \in \mathbb{R}^3$ corresponds to a 3D coordinate. Prior autoregressive approaches (Cheng et al., 2022; Sun et al., 2020; Chen et al., 2023) mainly follow the next-point prediction paradigm:

$$p(\mathbf{x}_1, \mathbf{x}_2, \ldots, \mathbf{x}_N) = \prod_{i=1}^N p(\mathbf{x}_i | \mathbf{x}_{i-1}, \ldots, \mathbf{x}_2, \mathbf{x}_1). \tag{1}$$

Training Eq. 1 requires a sequential classification objective. Each point $\mathbf{x}_i$ is first converted into a discrete integer token $q_i$ via VQ-VAE quantization (van den Oord et al., 2017), and the resulting tokens are flattened into a sequence $(q_1, \ldots, q_N)$ according to a predefined generation order. Autoregressive modeling is then applied to this discrete sequence following the paradigm in Eq. 1. However, this approach struggles to preserve permutation invariance, since the probability distribution depends on the chosen token ordering and is not invariant to different permutations of the points. In this work, instead of predicting the next point, we propose a novel autoregressive framework that predicts the next-scale level-of-detail (LoD) of the point cloud $\mathbf{X}$, while simultaneously preserving the permutation invariance property:

$$p(\pi(\mathbf{x}_1, \ldots, \mathbf{x}_N)) = p(\mathbf{x}_1, \ldots, \mathbf{x}_N), \qquad \forall \, \pi \in S_N. \tag{2}$$

In general, we construct $k$ different LoDs of $\mathbf{X}$, forming a coarse-to-fine sequence of global shapes $\mathbf{X}_1, \ldots, \mathbf{X}_K$, where each $\mathbf{X}_k \in \mathbb{R}^{s_k \times 3}$ represents a global shape at resolution $s_k$, obtained by

downsampling the original $\mathbf{X}$. Then, *PointNSP* is designed to learn the following distribution:

$$p(\mathbf{X}_1, \mathbf{X}_2, \ldots, \mathbf{X}_K) = \prod_{k=1}^{K} p(\mathbf{X}_k | \mathbf{X}_{k-1}, \ldots, \mathbf{X}_2, \mathbf{X}_1). \tag{3}$$

In this work, we set $s_1 = 1$ (a single starting point) and $s_K = N$ (total number of points). The generation process can be interpreted as an autoregressive upsampling procedure with sequential upsampling rates $r_1, r_2, \ldots, r_{K-1}$, satisfying the relationship $r_{K-1} \times \cdots \times r_1 \times s_1 = s_K = N$. Learning Eq. 3 requires a two-stage training process. The first stage involves training multi-scale VQ-VAE tokenizers to progressively reconstruct different LoDs, while the second stage trains the autoregressive transformer over the resulting multi-scale token sequence.

---

**Algorithm 1** Multi-scale VQVAE encoder

1: **Input:** 3D point cloud $\mathbf{X} = \{\mathbf{x}_i\}_{i=1}^{N}$.
2: **Hyperparameters:** # scales $\{s_k\}_{k=1}^{K}$.
3: $f = \mathcal{E}(\mathbf{X}), Q = [];$
4: **for** $k = 1, \cdots, K$ **do**
5: $\quad \mathbf{X}_k = \mathrm{FPS}(\mathbf{X}, s_k)$
6: $\quad q_k = \mathcal{Q}(f_k), f_k = \mathrm{Query}(f, \mathbf{X}_k)$
7: $\quad Q = \mathrm{queue\_push}(Q, q_k)$
8: $\quad \mathbf{z}_k = \mathrm{lookup}(Z, q_k)$
9: $\quad \mathbf{z}_k = \mathrm{upsampling}(\mathbf{z}_k, s_K)$
10: $\quad f = f - \phi_k(\mathbf{z}_k)$
11: **end for**
12: **Return:** token sequence $Q$.

---

**Algorithm 2** Multi-scale VQVAE decoder

1: **Input:** token sequence $Q$.
2: **Hyperparameters:** # scales $\{s_k\}_{k=1}^{K}$.
3: $\hat{f} = 0$
4: **for** $k = 1, \cdots, K$ **do**
5: $\quad q_k = \mathrm{queue\_pop}(Q)$
6: $\quad \mathbf{z}_k = \mathrm{lookup}(Z, q_k)$
7: $\quad \mathbf{z}_k = \mathrm{upsampling}(\mathbf{z}_k, s_K)$
8: $\quad \hat{f} = \hat{f} + \phi_k(\mathbf{z}_k)$
9: **end for**
10: $\hat{\mathbf{X}} = \mathcal{D}(\hat{f})$
11: **Return:** reconstructed point cloud $\hat{\mathbf{X}}$

---

### 3.2 MULTI-SCALE LoD REPRESENTATION

**Sampling Multi-Scale LoD Sequence.** To obtain $\mathbf{X}_1, \ldots, \mathbf{X}_K$, we employ the Farthest Point Sampling (FPS) (Eldar et al., 1997) algorithm to downsample the original point cloud $\mathbf{X}$. FPS offers two key advantages. First, it is permutation-invariant: point selection depends solely on spatial locations (geometric distances) rather than the order of points in the input array. The stochasticity of the downsampled point cloud arises only from the choice of the initial point, not from the input ordering. Second, FPS operates directly on the original point cloud, enabling straightforward querying of specific point embeddings via stored indices. Importantly, we do not fix the initial starting point of FPS, so the sampled points in $\mathbf{X}_k$ vary during training. This inherent stochasticity enhances model generalization, effectively serving as a form of data augmentation at each scale. Note that $\mathbf{X}_1, \ldots, \mathbf{X}_K$ are all obtained directly from the original point cloud $\mathbf{X}$, i.e., $\mathbf{X}_k = \mathrm{FPS}(\mathbf{X})$ for each scale $k$, rather than using an iterative downsampling strategy such as $\mathbf{X}_k = \mathrm{FPS}^k(\mathbf{X})$ adopted in 2D image VAR (Tian et al., 2024), where $\mathrm{FPS}^k$ denotes a nested operation with $k$ loops. Directly sampling points from $\mathbf{X}$ in this way increases the shape coverage at each scale of the LoD.

**Multi-Scale Feature Extraction.** For the encoder to obtain point features $f \in \mathbb{R}^{N \times 3}$ from the point cloud $\mathbf{X}$, any permutation-equivariant network $\mathrm{NN}(\cdot)$ is applicable: the per-point features reorder consistently with any permutation of the input points, i.e., $\pi(\mathrm{NN}(\mathbf{x}_1, ..., \mathbf{x}_N)) = \mathrm{NN}(\pi(\mathbf{x}_1, ..., \mathbf{x}_N))$ for any permutation $\pi$. Therefore, permutation-equivariant architectures such as PointNet (Qi et al., 2017a), PointNet++ (Qi et al., 2017b), PointNeXt (Qian et al., 2022) and PVCNN (Liu et al., 2019) are all applicable. To construct the multi-scale representations, we extract latent features $f_1 \in \mathbb{R}^{s_1 \times d}, \ldots, f_K \in \mathbb{R}^{s_K \times d}$ in a residual querying manner: each $\mathbf{X}_k$ retrieves features from the portion of $f$ not already captured by earlier scales, using the stored indices from downsampling. This ensures that later scales focus on complementary information rather than duplicating what has already been encoded. The resulting features are then used for multi-scale VQVAE codebook training.

**Multi-Scale RVQ Tokenizer.** We use residual vector quantization (RVQ) (Tian et al., 2024; Lee et al., 2022) to approximate $f$ progressively. Each residual feature $f_k$ is quantized into tokens $q_k = \{q_k^1, q_k^2, \ldots, q_k^{s_k}\} = \mathcal{Q}(f_k) \in [V]^{s_k}$, where each $q_k^i$ indexes the nearest codebook embedding

Figure 3: Illustration of training a multi-scale VQ-VAE for 3D point cloud reconstruction across scales $s_1$ to $s_4$, resulting in a multi-scale token sequence $Q = (q_1, \ldots, q_4)$.

$\mathbf{z}_v \in \mathbb{R}^d$: $q_k^i = \arg\min_{v \in [v]} \|\mathbf{z}_v - f_k^i\|$. This produces scale token sets $q_1, \ldots, q_K$ and corresponding embeddings $\mathbf{z}_1 \in \mathbb{R}^{s_1 \times d}, \ldots, \mathbf{z}_K \in \mathbb{R}^{s_K \times d}$, which together provide a residual approximation of the original feature $f$ as follows:

$$f_k = f_{k-1} - \phi_k(\text{upsampling}(\mathbf{z}_k, s_K)), \ f_0 = f \qquad (4)$$

Here $\phi_k(\cdot) : \mathbb{R}^{N \times d} \to \mathbb{R}^{N \times d}$ refines the latent embedding. The upsampling$(\cdot, s_K)$ module consists of a series of reshaping operations to upsample the latent $\mathbf{z}_k \in \mathbb{R}^{s_k \times d}$ to the highest (target) resolution $s_K \times d$. Compared to the vanilla VQ (van den Oord et al., 2017), the above RVQ mechanism allows finer approximation of the original $f$ with more efficient codebook utilization. We could compute the partial sum $\hat{f}$ and reconstruct the final predicted 3D shape $\hat{\mathbf{X}}$ as follows:

$$\hat{\mathbf{X}} = D(\hat{f}), \ \hat{f} = \sum_{k=1}^{K} \phi_k(\text{upsampling}(\mathbf{z}_k, s_K)), \ \mathbf{z}_k = \text{lookup}(q_k), \qquad (5)$$

where $D(\cdot) : \mathbb{R}^{N \times d} \to \mathbb{R}^{N \times 3}$ is a simple MLP decoder. Eq. 5 shows that $\hat{\mathbf{X}}$ is obtained by aggregating information across all scales of the level-of-detail (LoD). The pseudo-algorithms and process are illustrated in Algorithm 1, Algorithm 2, and Figure 3.

**Upsampling & Reconstruction.** The upsampling operation in both Eq. 4 and Eq. 5 follows a PU-Net (Yu et al., 2018) like operation, which consists of the duplicating and reshaping:

$$\mathbf{z}_k(s_k \times d) \xrightarrow{\text{duplicate}} \mathbf{z}_k(s_k \times r \times d) \xrightarrow{\text{reshape}} \mathbf{z}_k((s_k * r) \times d) = \mathbf{z}_K(s_K \times d), \qquad (6)$$

where the upsampling rate $r = \frac{s_K}{s_k}$ can be pre-computed. This upsampling operation can densify points by arbitrary factors while preserving the fundamental permutation-equivariance property. The reconstruction loss function to be minimized is defined as follows:

$$\mathcal{L}_{\text{recon}} = \mathcal{L}_{\text{CD}}(\mathbf{X}, \hat{\mathbf{X}}) + \mathcal{L}_{\text{EMD}}(\mathbf{X}, \hat{\mathbf{X}}) + \sum_{k=1}^{K} \|f[k] - sg(\mathbf{z}_k)\|_2^2 \qquad (7)$$

Here, $\mathcal{L}_{\text{CD}}$ denotes the Chamfer Distance loss, and $\mathcal{L}_{\text{EMD}}$ denotes the Earth-Mover Distance loss. These two terms are commonly used in prior works to measure the distance between point clouds from complementary perspectives. The stop-gradient operation, $sg[\cdot]$, ensures that the latent features $f$ used for reconstruction remain consistent with the quantized latent vectors $\mathbf{z}$.

## 3.3 AUTOREGRESSIVE TRANSFORMER FOR NEXT-SCALE LoD PREDICTION

The next step is to train an autoregressive transformer on the multi-scale token sequence $Q = (q_1, \ldots, q_K)$ obtained from the previous stage. Due to the strong local-geometry inductive bias inherent in 3D structures and the relatively smaller dataset size compared to 2D images, a naive causal transformer struggles to efficiently capture both intra-scale and inter-scale interactions. Consequently, it becomes necessary to embed geometric information into the transformer design, though this introduces additional challenges due to the unordered nature of point clouds.

Figure 4: Illustration of training a multi-scale 3D point cloud causal transformer with intermediate decoding, upsampling, position-aware soft masks, and block-wise causal masks.

**Inter-Scale Interaction Modeling.** Inter-scale interactions across all scales $(q_1, \ldots, q_K)$ are essential for the model to capture relationships between scales and to generate the next LoD conditioned on the preceding ones. The model is guided by a key principle: *tokens at scale $\hat{q}_k$ are restricted to attend only to tokens from the preceding scales $\hat{q}_1, \ldots, \hat{q}_{k-1}$. Within each scale, however, all tokens $q_k = (q_k^1, \ldots, q_k^{s_k})$ are allowed to fully attend to one another, ensuring that the model interprets them as complete shapes at the corresponding resolution.* To enforce this constraint, we construct the causal mask $\mathbf{M}$ as a block-diagonal matrix, where each diagonal block $\mathbf{M}_k$ of size $s_k \times s_k$ is fully unmasked: $\mathbf{M} = \text{diag}[\mathbf{M}_1, \mathbf{M}_2, ..., \mathbf{M}_K]$. This design allows bidirectional dependencies within each scale while preserving a lower-triangular structure across scales, thereby preventing information leakage from future tokens. To further distinguish scales, each token is assigned a scale embedding $\mathbf{s}_k^i \in \mathbb{R}^d$, implemented as a one-hot vector over the $K$ scales and shared among tokens within the same scale. Some algorithmic details are explained in the Appendix B.

**Intra-Scale Interaction Modeling.** Capturing intra-scale interactions within the token set $q_k = (q_k^1, \ldots, q_k^{s_k})$ is crucial for the model to effectively understand shapes across different resolutions. An important geometric prior is the relative distance between pairs of point tokens. Incorporating this information into intra-scale bidirectional attention allows tokens to place greater weight on nearby neighbors rather than starting from uniform attention. Specifically, we compute the position-aware soft masking matrix $\mathbf{M}_k^p \in \mathbb{R}^{s_k \times s_k}$, which is derived from the coordinate-based positional embedding matrix $\mathbf{P}_k \in \mathbb{R}^{s_k \times d}$, as follows:

$$\mathbf{M}_k^p = \text{Softmax}((\mathbf{P}_k \mathbf{W}_p)(\mathbf{P}_k \mathbf{W}_p)^T), \quad \mathbf{W}_p \in \mathbb{R}^{d \times d}. \tag{8}$$

$\mathbf{M}_k^p$ is a symmetric matrix, where each entry $\mathbf{M}_k^p[i, j] \in (0, 1)$ encodes the relative position between points $i$ and $j$. This raises a key question: *how can we derive the positional embedding $\mathbf{P}_k$ for each scale when no explicit 3D geometry is available at this stage?* Our solution is an intermediate-structure decoding strategy. Specifically, for each scale $k$, we apply the decoder $D(\cdot)$ to reconstruct the intermediate structure $\hat{\mathbf{X}}_{\mathbf{k-1}}$ using all tokens predicted up to step $k-1$:

$$\hat{\mathbf{X}}_{\mathbf{k-1}} = D(\sum_{m=1}^{k-1} \phi_m(\text{upsampling}(\hat{\mathbf{z}}_m, s_m))), \ \{\hat{\mathbf{z}}_1, \ldots, \hat{\mathbf{z}}_{k-1}\} = \text{lookup}(Z, \{\hat{q}_1, \ldots, \hat{q}_{k-1}\}). \tag{9}$$

$\hat{\mathbf{X}}_{\mathbf{k-1}} \in \mathbb{R}^{s_{k-1} \times 3}$ serves as the input coordinate information for the next scale $k$ prediction. The positional embedding is derived as an absolute positional encoding based on the 3D coordinates $\hat{\mathbf{X}}_{\mathbf{k-1}}$ using trigonometric functions (e.g. sin and cos). Constructing the encoding directly from 3D coordinates, rather than point indices, preserves permutation equivariance. The detailed derivation of $\mathbf{P}$ is provided in the Appendix B. Note that any positional encoding based on token indices should not be applied, as it would violate the permutation-equivariant property.

The prediction of each token $\hat{q}_k^i$ is evaluated using the cross-entropy (CE) loss $\mathcal{L}_k^i$. We first compute intra-scale loss $\mathcal{L}_k = \frac{1}{s_k} \sum_{i=1}^{s_k} \mathcal{L}_k^i$ and then compute inter-scale loss $\mathcal{L}_{\text{total}} = \frac{1}{K} \sum_{k=1}^{K} \mathcal{L}_k$. The second stage training architecture is illustrated in Figure 4. We provide a theoretical analysis of the permutation invariance of *PointNSP* 's distribution modeling in Appendix A.

Table 1: The *Performance* (1-NNA) is evaluated based on single-class generation. The second block specifies the types of generative models used in each study. The best performance is highlighted in **bold**, while the second-best performance is underlined. Performance is reported on two dataset splits: the top corresponds to the random split, and the bottom corresponds to the LION split.

| Model | Generative Model | Airplane | | Chair | | Car | | Mean CD ↓ | Mean EMD ↓ |
|---|---|---|---|---|---|---|---|---|---|
| | | CD ↓ | EMD ↓ | CD ↓ | EMD ↓ | CD ↓ | EMD ↓ | | |
| 1-GAN Achlioptas et al. (2018) | GAN | 87.30 | 93.95 | 68.58 | 83.84 | 66.49 | 88.78 | 74.12 | 88.86 |
| PointFlow (Yang et al., 2019) | Normalizing Flow | 75.68 | 70.74 | 62.84 | 60.57 | 58.10 | 56.25 | 65.54 | 62.52 |
| DPF-Net (Klokov et al., 2020) | Normalizing Flow | 75.18 | 65.55 | 62.00 | 58.53 | 62.35 | 54.48 | 66.51 | 59.52 |
| SoftFlow (Kim et al., 2020) | Normalizing Flow | 76.05 | 65.80 | 59.21 | 60.05 | 64.77 | 60.09 | 66.67 | 61.98 |
| SetVAE (Kim et al., 2021) | VAE | 75.31 | 77.65 | 58.76 | 61.48 | 59.66 | 61.48 | 64.58 | 66.87 |
| ShapeGF (Cai et al., 2020) | Diffusion | 80.00 | 76.17 | 68.96 | 65.48 | 63.20 | 56.53 | 70.72 | 66.06 |
| DPM (Luo & Hu, 2021) | Diffusion | 76.42 | 86.91 | 60.05 | 74.77 | 68.89 | 79.97 | 68.45 | 80.55 |
| PVD-DDIM (Song et al., 2021a) | Diffusion | 76.21 | 69.84 | 61.54 | 57.73 | 60.95 | 59.35 | 66.23 | 62.31 |
| PSF (Wu et al., 2023b) | Diffusion | 74.45 | 67.54 | 58.92 | 54.45 | 57.19 | 56.07 | 62.41 | 57.20 |
| PVD (Zhou et al., 2021) | Diffusion | 73.82 | 64.81 | 56.26 | 53.32 | 54.55 | 53.83 | 61.54 | 57.32 |
| LION (Zeng et al., 2022) | Diffusion | 72.99 | 64.21 | 55.67 | 53.82 | 53.47 | 53.21 | 61.75 | 57.59 |
| DIT-3D (Mo et al., 2023) | Diffusion | - | - | 54.58 | 53.21 | - | - | - | - |
| TIGER (Ren et al., 2024) | Diffusion | 73.02 | 64.10 | 55.15 | 53.18 | 53.21 | 53.95 | 60.46 | 57.08 |
| PointGrow (Sun et al., 2020) | Autoregressive | 82.20 | 78.54 | 63.14 | 61.87 | 67.56 | 65.89 | 70.96 | 68.77 |
| CanonicalVAE (Cheng et al., 2022) | Autoregressive | 80.15 | 76.27 | 62.78 | 61.05 | 63.23 | 61.56 | 68.72 | 66.29 |
| PointGPT (Chen et al., 2023) | Autoregressive | 74.85 | 65.61 | 57.24 | 55.01 | 55.91 | 54.24 | 63.44 | 62.24 |
| *PointNSP*-s (*ours*) | Autoregressive | 72.92 | 63.98 | 54.89 | 53.02 | 52.86 | 52.07 | 60.22 | 56.36 |
| *PointNSP*-m (*ours*) | **Autoregressive** | **72.24** | **63.69** | **54.54** | **52.85** | **52.17** | **51.85** | **59.65** | **56.13** |
| LION | Diffusion | 67.41 | 61.23 | 53.70 | 52.34 | 53.41 | 51.14 | 58.17 | 54.90 |
| TIGER | Diffusion | 67.21 | 56.26 | 54.32 | 51.71 | 54.12 | 50.24 | 58.55 | 52.74 |
| *PointNSP*-s (*ours*) | Autoregressive | 67.15 | 56.12 | 54.22 | 51.19 | 53.98 | 50.15 | 58.45 | 52.49 |
| *PointNSP*-m (*ours*) | **Autoregressive** | **66.98** | **56.05** | **54.01** | **53.76** | **53.12** | **50.09** | **58.04** | **52.30** |

## 4 EXPERIMENTS

### 4.1 EXPERIMENTAL SETUP

**Datasets.** In line with prior studies, we use ShapeNetv2, pre-processed by Point-Flow (Yang et al., 2019), as our primary dataset. Following previous experimental setups, we focus on the three categories with the largest number of samples: airplanes, chairs, and cars. Each shape is globally normalized and sampled to 2048

Table 2: Training time (in GPU hours, averaged over three categories), sampling time (in seconds, averaged over 10 samples), and model size (in millions of parameters). Ranked by MeanCD and MeanEMD.

| Quality Rank | Model | Training Time (GPU hours) ↓ | Sampling Time (s) ↓ | Param (M) ↓ |
|---|---|---|---|---|
| 8 | PointGrow | 156 | 5.80 | 25 |
| 7 | CanonicalVAE | 142 | 5.45 | 30 |
| 6 | PointGPT | 185 | 5.32 | 46 |
| 5 | PVD | 142 | 29.9 | 45 |
| 4 | LION | 550 | 31.2 | 60 |
| 3 | TIGER | 164 | 23.6 | 55 |
| 2 | *PointNSP*-s (*ours*) | **125** | **3.21** | **22** |
| 1 | *PointNSP*-m (*ours*) | 178 | 3.59 | 32 |

points. The training set comprises 2832 airplanes, 4612 chairs, and 2458 cars, while the evaluation set contains 405 airplanes, 662 chairs, and 352 cars. Experiments are conducted on two data splits inherited from LION (Zeng et al., 2022) and TIGER (Ren et al., 2024).

**Metrics.** In line with the benchmarks established by PVD (Zhou et al., 2021) and LION (Zeng et al., 2022), we adopt the 1-NN (1-nearest neighbor) accuracy (Lopez-Paz & Oquab, 2017) as our evaluation criterion. This metric has demonstrated its effectiveness in assessing both the quality and diversity of the generated point clouds, with a score approaching 50% indicating exceptional performance (Yang et al., 2019). To calculate the 1-NN distance matrix, we employ two widely recognized metrics for measuring the distance between point clouds: Chamfer Distance (CD) and Earth Movers' Distance (EMD). We also present the mean CD and the mean EMD by calculating the average CD and EMD across three distinct categories. To assess efficiency, we evaluate the training time by documenting the GPU hours and gauge the sampling efficiency by reporting the average inference time over 10 randomly generated samples.

### 4.2 PERFORMANCE COMPARISON FOR SINGLE-CLASS GENERATION

**Results.** To evaluate the performance of *PointNSP*, we compare it against several strong baseline models. Specifically, we include state-of-the-art (SoTA) diffusion-based methods: DPM (Luo & Hu, 2021), PVD (Zhou et al., 2021), LION (Zeng et al., 2022), DIT-3D (Mo et al., 2023), and TIGER (Ren et al., 2024). We also evaluate PVD-DDIM (Song et al., 2021a), which is designed to accelerate the sampling process of PVD. In addition, we include three leading autoregressive models: PointGrow (Sun et al., 2020), CanonicalVAE (Cheng et al., 2022), and PointGPT (Chen et al., 2023). We report results for two variants of our model—*PointNSP*-s and *PointNSP*-m—representing small and medium parameter configurations, respectively. Note that we only compare against models with publicly available code. **Quality.** As shown in Table 1, *PointNSP* consistently outperforms all baseline models on both the conventional and LION data splits, achieving state-of-the-art generation quality in terms of both CD and EMD metrics. Notably, even the lightweight *PointNSP*-s demonstrates

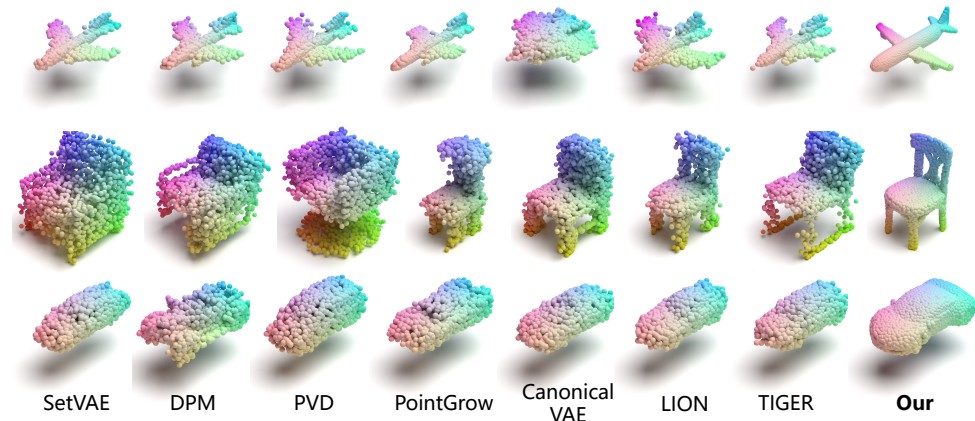

Figure 5: Our generation results (right) compared to baseline models (left). *PointNSP* generates high-quality and diverse 3D point clouds.

Table 3: Ablation studies on the Transformer backbones, upsampling networks, token embedding, and FPS stochasticity.

| Mask | Upsampling | | FPS Stochasticity | TE | Mean CD ↓ | Mean EMD ↓ |
|---|---|---|---|---|---|---|
| | Voxel | PU-Net | | | | |
| | | ✓ | | SE | 64.25 | 60.53 |
| ✓ | | ✓ | | SE+A-PE | 62.19 | 58.23 |
| ✓ | ✓ | | | SE+A-PE | 63.05 | 58.47 |
| ✓ | | ✓ | | L-PE | 63.22 | 59.71 |
| ✓ | | ✓ | | A-PE | 62.12 | 60.02 |
| ✓ | | ✓ | ✓ | A-PE | 61.28 | 57.32 |
| ✓ | | ✓ | ✓ | SE+L-PE | 60.62 | 57.34 |
| ✓ | | ✓ | ✓ | **SE+A-PE** | **59.65** | **56.13** |

Table 4: Multi-Class Generation results (1-NNA↓) on ShapeNet dataset from PointFlow. All data normalized individually into [-1, 1].

| | Airplane | | Chair | | Car | |
|---|---|---|---|---|---|---|
| | CD ↓ | EMD ↓ | CD ↓ | EMD ↓ | CD ↓ | EMD ↓ |
| PVD | 97.53 | 99.88 | 88.37 | 96.37 | 89.77 | 94.89 |
| TIGER | 83.54 | 81.55 | 57.34 | 61.45 | 65.79 | 57.24 |
| PointGPT | 94.94 | 91.73 | 71.83 | 79.00 | 89.35 | 87.22 |
| LION | 86.30 | 77.04 | 66.50 | 63.85 | 64.52 | 54.21 |
| *PointNSP*-s (*ours*) | 78.95 | 68.84 | 58.79 | 55.10 | 59.97 | 52.89 |
| *PointNSP*-m (*ours*) | **75.42** | **66.54** | **56.03** | **52.22** | **57.95** | **49.55** |

competitive performance. **Efficiency.** Table 2 presents the training time, sampling time, and number of parameters for several strong baseline models from Table 1, including PVD, LION, TIGER, and PointGPT. In terms of training efficiency, *PointNSP*-s achieves the shortest training time among all selected methods, while *PointNSP*-m demonstrates comparable training costs to state-of-the-art diffusion-based approaches. For inference efficiency, both *PointNSP*-s and *PointNSP*-m outperform selected strong baseline methods with significantly faster sampling speeds. This advantage is primarily due to *PointNSP*'s parallel token generation within each scale. Additionally, *PointNSP* exhibits the highest parameter efficiency among all compared approaches. Notably, *PointNSP*-s has a smaller parameter size than PointGrow, despite the latter performing significantly worse in generation quality. Compared to diffusion-based methods, *PointNSP*-m offers a substantial advantage in parameter efficiency, requiring nearly half the number of parameters while maintaining superior performance.

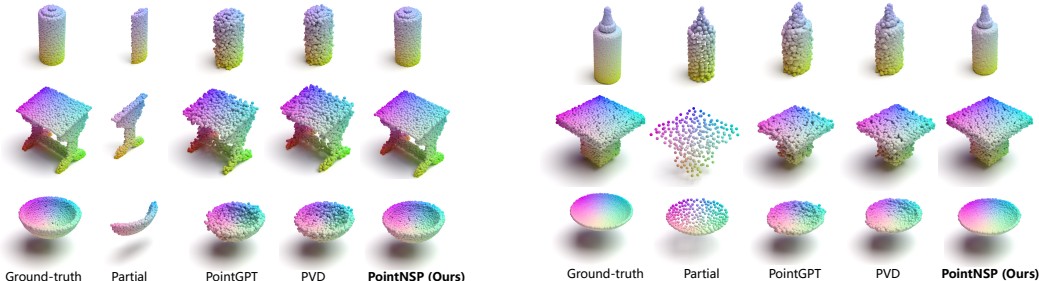

Figure 6: (Left) Visualizations of point cloud completion results. (Right) Visualizations of point cloud upsampling results.

### 4.3 ABLATION AND ANALYSIS

**Neural Architectures & Training Strategies** We conduct comprehensive ablation studies to assess the impact of various architectural components and training strategies. First, we compare two upsampling strategies: *voxel-based representations* and *PU-Net*. Our experiments show that PU-Net consistently outperforms voxel-based upsampling, owing to its permutation-equivariant design. Second, we evaluate the effectiveness of the *position-aware masking strategy*, which significantly

boosts model performance. Third, we analyze the contribution of each embedding module in the token embedding (TE) layer, with results highlighting the notable impact of the scale embedding. We also compare *learnable positional encoding (L-PE)* with *absolute PE (A-PE)*, finding that absolute PE yields better performance. Finally, we investigate the effect of FPS *stochasticity*, arising from the inherent randomness of the FPS algorithm. We compare against a variant with a fixed starting point, which produces deterministic downsampling. The results demonstrate that FPS stochasticity consistently enhances model generalization. These findings are summarized in Table 3, with additional ablation results—such as the effect of varying the number of scales—provided in the Appendix C.

### 4.4 Multi-class Generation & Dense Point Cloud Generation

Beyond the single-class generation setting, we further evaluate all models on the more challenging multi-class generation task introduced by LION (Zeng et al., 2022), aiming to assess their ability to generalize across diverse object categories. Specifically, we train *PointNSP* and all baseline models using class conditioning over 55 distinct categories from ShapeNet. This conditional setup allows the model to capture class-specific characteristics while also leveraging shared geometric patterns across categories. As shown in Table 4, *PointNSP* significantly outperforms all strong baselines. We provide visualizations of diverse shape classes in Appendix E. We further evaluate all models on high-resolution point cloud generation with up to 8192 points. Empirical results demonstrate that *PointNSP* maintains superior performance across all major metrics, with a widening performance margin as resolution increases—particularly training and inference speed (Results are shown in Table 5 (right) and the Appendix D).

### 4.5 Point Cloud Completion & Upsampling

We evaluate *PointNSP* on two key downstream tasks: point cloud completion and upsampling. For completion, we follow the experimental setup of PVD (Zhou et al., 2021). For upsampling, we use a factor of $r = 2$, increasing the input point clouds from 1024 to 2048 points. As shown in Table 5 (Left & Middle), *PointNSP* consistently achieves the best performance on point cloud completion. For the upsampling task, it outperforms all selected baselines across both metrics, highlighting its effectiveness on diverse downstream applications and its potential as a foundation model. Visualization results for these two tasks are presented in Figure 6 and Appendix E.

Table 5: (Left) The shape completion task in PVD; (Middle) Our own upsampling task by upsampling to denser point clouds; (Right) Denser point cloud generation with 8192 points.

| Category | Model | CD↓ | EMD↓ | Category | Model | CD↓ | EMD↓ | Category | Model | CD↓ | EMD↓ |
|---|---|---|---|---|---|---|---|---|---|---|---|
| | SoftFlow | 40.42 | 11.98 | | PVD | 73.56 | 71.65 | | PVD | 69.77 | 69.98 |
| | PointFlow | 40.30 | 11.80 | | TIGER | 71.65 | 59.94 | | TIGER | 68.48 | 60.24 |
| Airplane | DPF-Net | 52.79 | 11.05 | Airplane | PointGPT | 72.11 | 60.12 | Airplane | PointGPT | 69.29 | 61.56 |
| | PVD | 44.15 | 10.30 | | LION | 70.41 | 59.65 | | LION | 68.95 | 60.38 |
| | *PointNSP*-m (*ours*) | **40.12** | **10.08** | | *PointNSP*-m (*ours*) | **68.89** | **58.86** | | *PointNSP*-m (*ours*) | **66.63** | **55.29** |
| | SoftFlow | 27.86 | 32.95 | | PVD | 53.81 | 64.61 | | PVD | 52.56 | 51.33 |
| | PointFlow | 27.07 | 36.49 | | TIGER | 52.80 | 52.98 | | TIGER | 51.87 | 51.85 |
| Chair | DPF-Net | 27.63 | 33.20 | Chair | PointGPT | 53.75 | 53.21 | Chair | PointGPT | 52.43 | 52.09 |
| | PVD | 32.11 | 29.39 | | LION | 53.98 | 54.33 | | LION | 53.76 | 53.45 |
| | *PointNSP*-m (*ours*) | **27.02** | **28.78** | | *PointNSP*-m (*ours*) | **52.04** | **51.03** | | *PointNSP*-m (*ours*) | **50.98** | **50.45** |
| | SoftFlow | 18.50 | 27.89 | | PVD | 58.95 | 48.43 | | PVD | 54.19 | 46.55 |
| | PointFlow | 18.03 | 28.51 | | PointGPT | 57.26 | 47.85 | | PointGPT | 54.97 | 45.20 |
| Car | DPF-Net | 13.96 | 23.18 | Car | TIGER | 57.90 | 48.01 | Car | TIGER | 53.78 | 44.12 |
| | PVD | 17.74 | 21.46 | | LION | 57.14 | 47.56 | | LION | 54.98 | 44.67 |
| | *PointNSP*-m (*ours*) | **13.84** | **20.68** | | *PointNSP*-m (*ours*) | **55.85** | **46.74** | | *PointNSP*-m (*ours*) | **52.40** | **43.05** |

## 5 Conclusions

In this study, we present *PointNSP*, a novel autoregressive framework for high-quality 3D point cloud generation. Unlike prior methods, *PointNSP* adopts a coarse-to-fine strategy that captures the multi-scale LoD of 3D shapes without reducing them to sequences of local predictions, thereby preserving global structural coherence throughout generation. Our approach consistently outperforms existing point cloud generative models in quality while delivering significant improvements in parameter efficiency, training efficiency, and inference speed. Looking ahead, we plan to scale *PointNSP* toward foundation-level models on large datasets and explore its integration with diffusion-based approaches as a promising future direction.

REPRODUCIBILITY STATEMENT

We include the implementation of our approach in the supplementary materials, with full hyper-parameter specifications documented in the Appendix C and embedded within the released code. Upon acceptance, we plan to publicly release both the source code and pretrained model checkpoints, accompanied by a dedicated project webpage for long-term maintenance.

LARGE LANGUAGE MODEL USAGE DECLARATION

In this work, ChatGPT OpenAI (2023) was used solely for polishing the wording of certain paragraphs and for verifying specific conceptual details.

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

# APPENDIX

## A  PERMUTATION INVARIANCE OF THE PROBABILITY DISTRIBUTION

Here we show why the distribution $p(\mathbf{X}_1, \mathbf{X}_2, \ldots, \mathbf{X}_K) = \prod_{k=1}^{K} p(\mathbf{X}_k | \mathbf{X}_{k-1}, \ldots, \mathbf{X}_2, \mathbf{X}_1)$ in Eq 3 preserves the permutation-invariance $p(\pi(\mathbf{x}_1, \ldots, \mathbf{x}_N)) = p(\mathbf{x}_1, \ldots, \mathbf{x}_N), \forall\, \pi \in S_N$ in Eq 2.

By definition, the joint distribution factorizes as:

$$p(\mathbf{X}_1, \mathbf{X}_2, \ldots, \mathbf{X}_K) = p(\mathbf{X}_1)p(\mathbf{X}_2|\mathbf{X}_1) \ldots p(\mathbf{X}_K|\mathbf{X}_1, \ldots \mathbf{X}_{K-1}). \tag{10}$$

Suppose the feature encoder and the conditional model for each scale are permutation-equivariant or permutation-invariant. Then, for any permutation $\pi_k$ of points within $\mathbf{X}_k$:

$$p(\pi_k(\mathbf{X}_k)|\mathbf{X}_1, \ldots \mathbf{X}_{k-1}) = p(\mathbf{X}_k|\mathbf{X}_1, \ldots \mathbf{X}_{k-1}). \tag{11}$$

This holds for each $k = 1, \ldots, K$.

Consider an arbitrary permutation $\pi$ acting on the full set of points $\mathbf{X} = \bigcup_{k=1}^{K} \mathbf{X}_k$. This permutation can be decomposed into independent permutations within each scale:

$$\pi = (\pi_1, \pi_2, \ldots, \pi_K), \qquad \pi_k \in S_{s_k}. \tag{12}$$

Then the joint distribution under this permutation is

$$
\begin{aligned}
p(\pi(\mathbf{X}_1, \mathbf{X}_2, \ldots, \mathbf{X}_K)) &= \prod_{k=1}^{K} p(\pi_k(\mathbf{X}_k)|\mathbf{X}_{k-1}, \ldots, \mathbf{X}_2, \mathbf{X}_1) \\
&= \prod_{k=1}^{K} p(\mathbf{X}_k|\mathbf{X}_{k-1}, \ldots, \mathbf{X}_2, \mathbf{X}_1) \ \text{(by permutation invariance at each scale)} \\
&= p(\mathbf{X}_1, \mathbf{X}_2, \ldots, \mathbf{X}_K)
\end{aligned}
\tag{13}
$$

The core operations in *PointNSP* —such as the feature encoder, upsampling, and FPS—are all designed to be permutation-equivariant or permutation-invariant. As a result, the autoregressive factorization preserves permutation invariance: permuting the points within any scale does not alter the joint probability.

$$p(\pi(\mathbf{x}_1, \ldots, \mathbf{x}_N)) = p(\mathbf{x}_1, \ldots, \mathbf{x}_N), \forall\, \pi \in S_N. \tag{14}$$

## B  ALGORITHM DETAILS

**Coordinate-based Positional Encoding.**  We adopt the absolute positional encoding strategy purely based on 3D coordinates used in TIGER (Ren et al., 2024). Based on our experiments, we find the Base $\lambda$ Position Encoding (B$\lambda$PE) performs better and here we present its formula:

$$p = \lambda^2 * z_i + \lambda * y_i + x_i \tag{15}$$

$$\mathbf{P}_k(p, 2i) = \sin\left(\frac{p}{10000^{\frac{2i}{D}}}\right) \tag{16}$$

$$\mathbf{P}_k(p, 2i+1) = \cos\left(\frac{p}{10000^{\frac{2i}{D}}}\right), \tag{17}$$

where $\mathbf{x}_i = (x_i, y_i, z_i) \in \mathbb{R}^3$, $p$ is a polynomial expression with hyperparameter coefficient $\lambda$. We set $\lambda = 1000$ following the setting in TIGER (Ren et al., 2024), which means this preserves three decimal places of precision. Here $\mathbf{P}_k \in \mathbb{R}^{s_k \times D}$ denotes the positional embedding of all tokens within the scale $k$. In short, we apply the B$\lambda$PE embedding strategy scale-by-scale at this time.

**Intra-Scale Token Embedding Layer.**  In this way, we derive the absolution positional encoding $\mathbf{p}_k^i$ for each token $q_k^i$ with predicted $\hat{\mathbf{X}}_k$. Additionally, the model needs to know which scale that each token belongs to. Therefore, $\mathbf{s}_k$ is a simple one-hot encoding $\mathbf{s}_k = \text{one-hot}(k)$ out of total $K$ scales. Tokens $(q_k^i, q_k^j)$ from the same scale $k$ share the same scale embedding $\mathbf{s}_k$ ($\mathbf{s}_k^i = \mathbf{s}_k^j$). Following the implementation of Llama 3 (Dubey et al., 2024), we add both the positional embedding $\mathbf{p}_k^i$ and the

scale embedding $\mathbf{s}_k^i$ to the query $\mathbf{u}_k^i$ and key vectors $\mathbf{v}_k^i$ derived in the attention mechanism. Then the token embedding operation for each token $q_k^i$ is as follows:

$$\mathbf{u}_k^i = \mathbf{W_U}\mathbf{z}_k^i + \mathbf{p}_k^i + \mathbf{s}_k^i, \quad \mathbf{v}_k^i = \mathbf{W_V}\mathbf{z}_k^i + \mathbf{p}_k^i + \mathbf{s}_k^i, \tag{18}$$

where $\mathbf{W_U}$ and $\mathbf{W_V}$ are projection matrices for queries and keys respectively. Then the queries and keys will be upsampled to $\mathbf{U}_k, \mathbf{V}_k = \mathrm{upsampling}(\mathbf{U}_k, \mathbf{V}_k) \in \mathbb{R}^{s_{k+1} \times d}$ following the Eq. 6, which will be fed to the transformer for next-scale token prediction.

## C   Hyperparameters and Reproducibility Settings

**Hyper-parameters and Reproducibility Settings.**   We mainly follow the setting in (Zeng et al., 2022). Specifically, we set the learning rate $3e^{-4}$ and the batch-size 32. We perform all the experiments on a workstation with Intel Xeon Gold 6154 CPU (3.00GHz) and 8 NVIDIA Tesla V100 (32GB) GPUs.

We use an AdamW optimizer with an initial learning rate of $10^{-4}$ for VQVAE training and $10^{-3}$ for autoregressive transformer training respectively. For up-sampling and completion experiments, we follow the experimental settings of PVD (Zhou et al., 2021).

| Hyperparameter | Value |
|---|---|
| # PVCNN layers | 4 |
| # PVCNN hidden dimension | 1024 |
| # PVCNN voxel grid size | 32 |
| # MLP layers | 6 |
| # Attention dimension | 1024 |
| # Attention head | 32 |
| Optimizer | AdamW |
| Weight Decay | 0.01 |
| LR Schedule | Cosine |

**The effect of # scales $K$.**   Figure S1 illustrates the impact of the total number of scales on the overall performance of *PointNSP*. As the number of scales increases, *PointNSP*'s performance improves accordingly. However, beyond $K = 11$ scales, no further performance gains are observed. We hypothesize that additional scales may require higher point cloud densities to be effective. Moreover, increasing the number of scales naturally leads to longer sampling times.

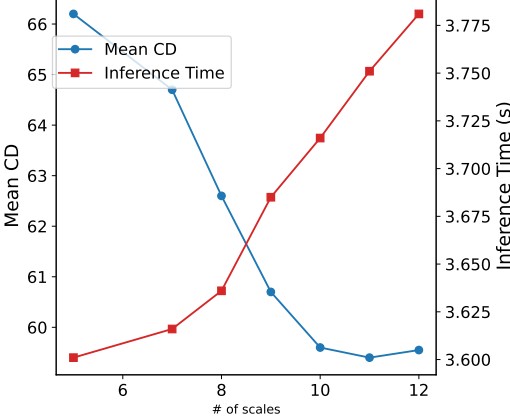

Figure S1: The effect of number of scales on the overall performance of *PointNSP*.

## D  MORE RESULTS ON DENSE POINT CLOUD GENERATION WITH 8192 POINTS

We show sampling time and training hours of different methods in Table S1. We could clearly see that *PointNSP* significantly outperforms other strong methods in terms of learning efficiency and inference efficiency, which shows its strong scalability potential.

Table S1: Training time (in GPU hours), sampling time (in seconds) on 8192 points. The reported training time is averaged across three categories: airplane, chair, and car. The sampling time is averaged over 10 random generations.

| Quality Rank | Model | Training Time (GPU hours) ↓ | Sampling Time (s) ↓ |
|---|---|---|---|
| 8 | PointGrow | 240 | 9.90 |
| 7 | CanonicalVAE | 205 | 8.95 |
| 6 | PointGPT | 296 | 10.56 |
| 5 | PVD | 201 | 58.1 |
| 4 | LION | 610 | 59.5 |
| 3 | TIGER | 320 | 42.1 |
| 2 | *PointNSP*-s (*ours*) | **175** | **4.54** |
| 1 | *PointNSP*-m (*ours*) | 190 | 5.48 |

## E  MORE VISUALIZATION RESULTS

**3D Point Clouds Generated by *PointNSP*.**    We showcase diverse 3D point clouds generated by *PointNSP*, covering a wide range of shapes (Figures S2 and S3). Additional results for single-class generation across five categories are presented in Figures S4–S8. We further illustrate the multi-scale sequential generation process in Figures S9–S11, and provide visualizations of point cloud completion and upsampling in Figure S12.

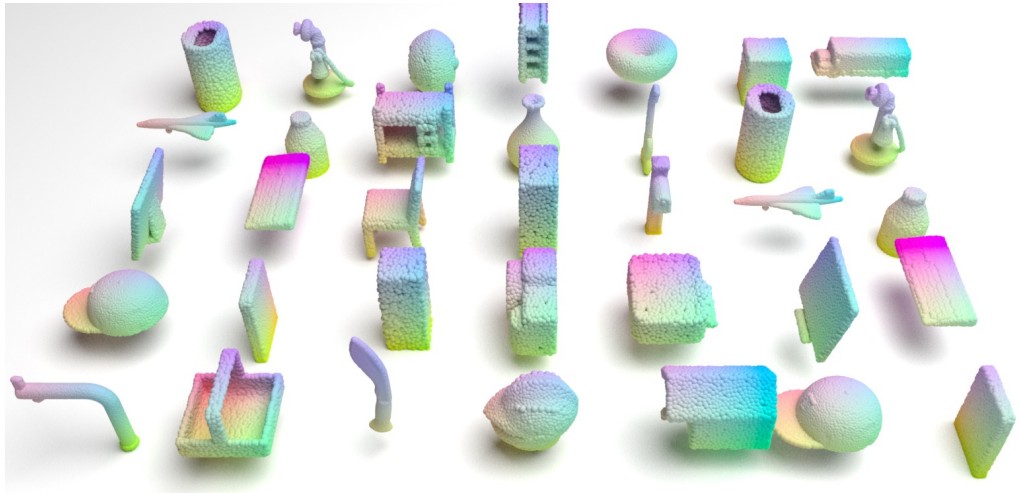

Figure S2: Generated shapes from the *PointNSP* model trained on ShapeNet's other categories.

## F  OTHER RELATED WORKS

**Point Cloud Upsampling.**    Point cloud upsampling is a crucial process in 3D modeling, aimed at increasing the resolution of low-resolution 3D point clouds. PU-Net (Yu et al., 2018) pioneered the use of deep neural networks for this task, laying the foundation for subsequent advancements. Models such as PU-GCN (Qian et al., 2021a) and PU-Transformer (Qiu et al., 2022) have further refined point cloud feature extraction by leveraging graph convolutional networks and transformer networks, respectively. Additionally, approaches like Dis-PU (Li et al., 2021), PU-EVA (Luo et al., 2021), and MPU (Du et al., 2022) have enhanced the PU-Net pipeline by incorporating cascading refinement architectures. Other methods, such as PUGeo-Net (Qian et al., 2020), NePs (Feng et al.,

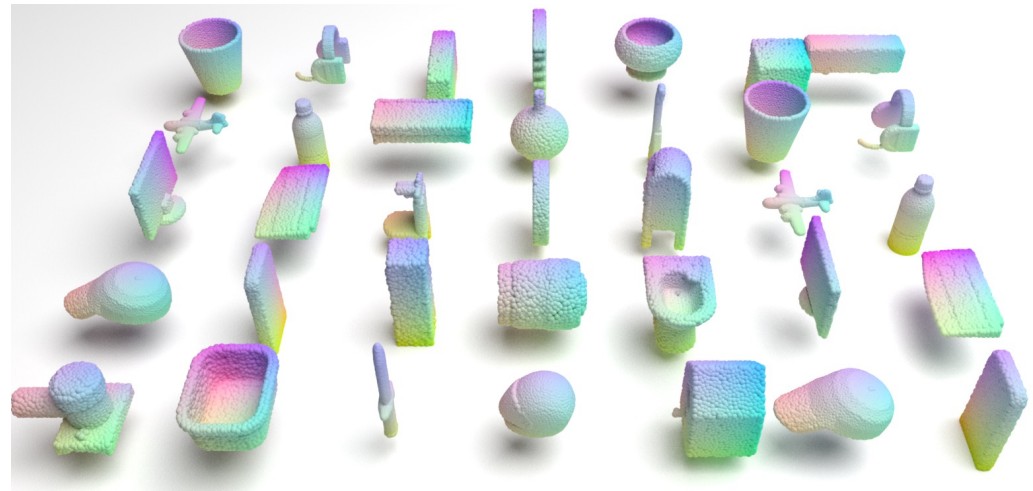

Figure S3: Generated shapes from the *PointNSP* model trained on ShapeNet's other categories.

2022), and MAFU (Qian et al., 2021b), employ local geometry projections into 2D space to model the underlying 3D surface. More recent approaches have reframed up-sampling as a generation task. For instance, PU-GAN (Li et al., 2019) and PUFA-GAN (Liu et al., 2022) leverage generative adversarial networks (GANs) to produce high-resolution point clouds. Grad-PU (He et al., 2023) first generates coarse dense point clouds through nearest-point interpolation and then refines them iteratively using diffusion models. In contrast, PUDM (Qu et al., 2024) directly utilizes conditional diffusion models, treating sparse point clouds as input conditions for generating denser outputs. In this work, our generative model, *PointNSP*, incorporates upsampling networks in both training stages, making it well-suited for enhancing downstream point cloud up-sampling tasks.

**Autoregressive Generative Models.** Autoregressive modeling has demonstrated exceptional success in natural language processing, leading to the development of powerful large language models (LLMs) such as (Sutskever et al., 2014; OpenAI, 2024; 2023; Patel et al., 2023). Building on the success of LLMs, researchers have explored autoregressive approaches across a wide range of domains, including image generation (van den Oord et al., 2017; Esser et al., 2021; Li et al., 2024; Tian et al., 2024; Lee et al., 2022; Chang et al., 2022), graph generation (You et al., 2018), video synthesis (Weissenborn et al., 2020), molecule generation (Shi et al., 2020; Schwaller et al., 2019), and protein sequence modeling (Madani et al., 2023; Lin et al., 2022). At its core, autoregressive modeling involves sequentially predicting future elements based on past outputs, making it particularly effective for audio generation tasks, where such temporal dependencies are naturally present.

## G  LIMITATIONS & BROADER IMPACT

*PointNSP* does not present major limitations, though the main challenge lies in learning high-quality multi-scale codebook embeddings for 3D point cloud representations—a process that remains resource-intensive. Several promising directions emerge for future work. One is scaling generation to produce ultra-dense point clouds (e.g., 10k–100k points), which could then be converted into high-quality meshes. Another is enabling fine-grained control over local geometric details, an essential step toward practical applications. While this work does not pose immediate societal risks, potential misuse for generating harmful 3D content warrants vigilance from the broader community. From a research perspective, *PointNSP* makes a meaningful contribution to both generative modeling and 3D point cloud research.

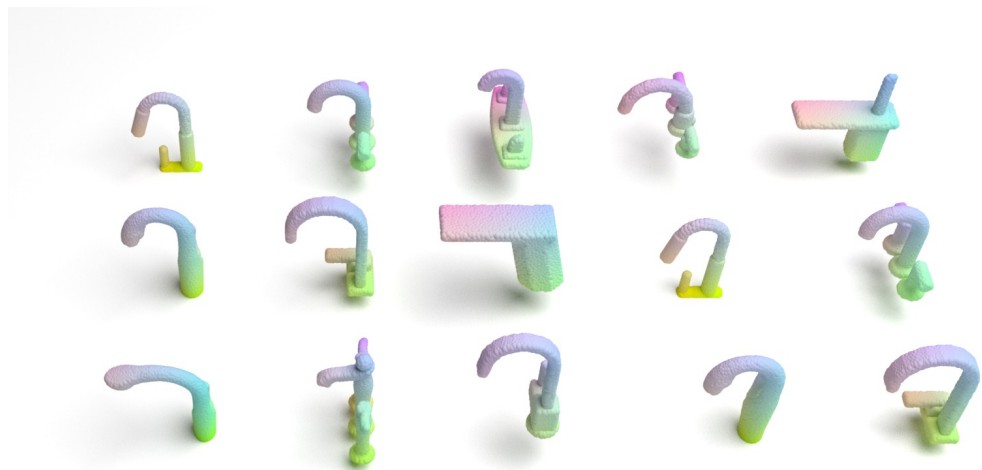

Figure S4: Generated single-class shapes from the *PointNSP* model trained on ShapeNet's other categories.

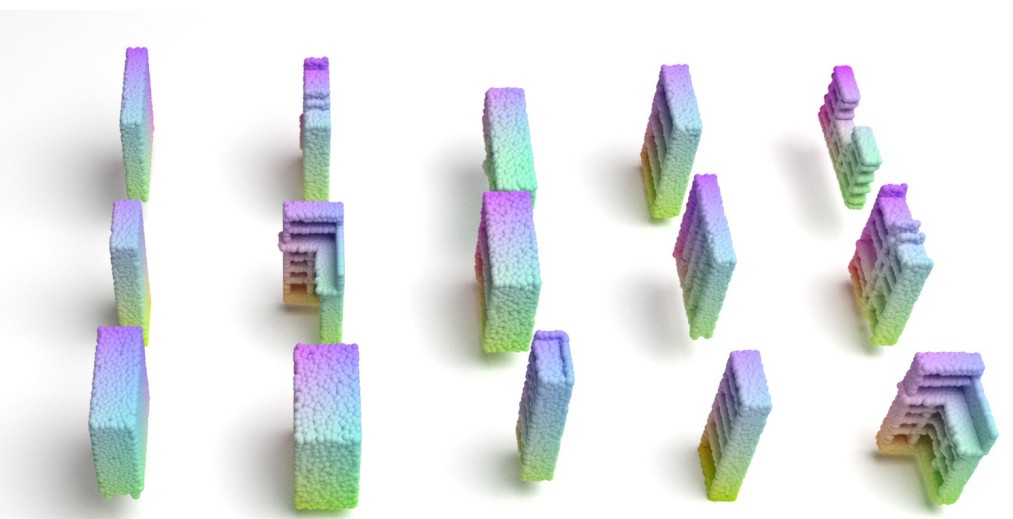

Figure S5: Generated single-class shapes from the *PointNSP* model trained on ShapeNet's other categories.

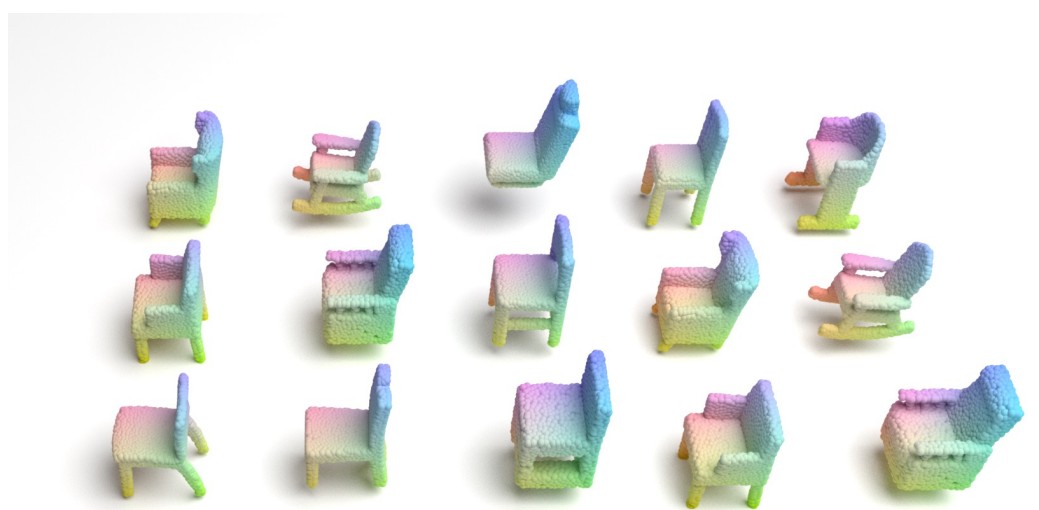

Figure S6: Generated single-class shapes from the *PointNSP* model trained on ShapeNet's other categories.

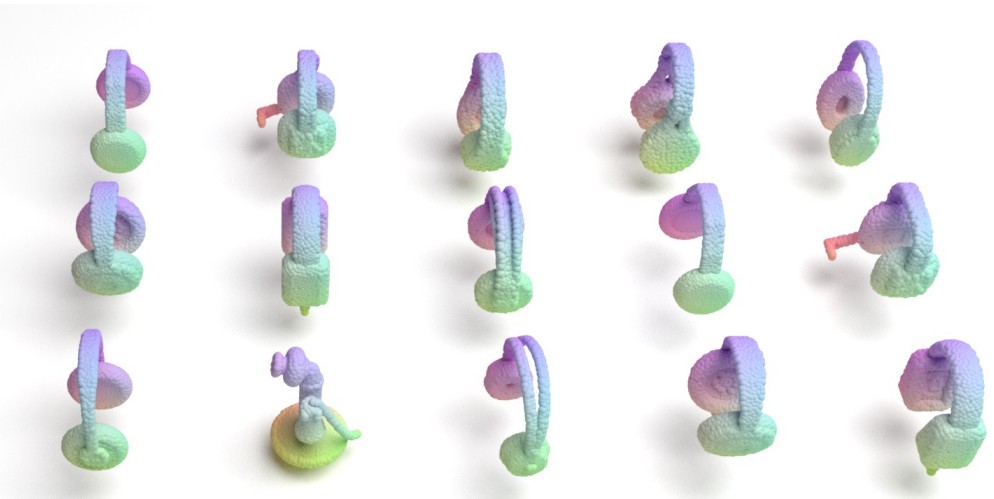

Figure S7: Generated single-class shapes from the *PointNSP* model trained on ShapeNet's other categories.

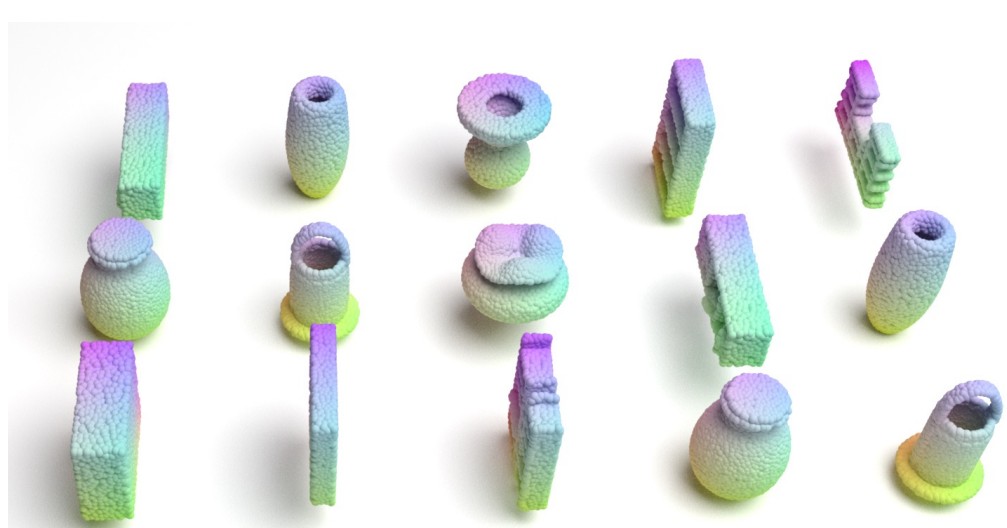

Figure S8: Generated single-class shapes from the *PointNSP* model trained on ShapeNet's other categories.

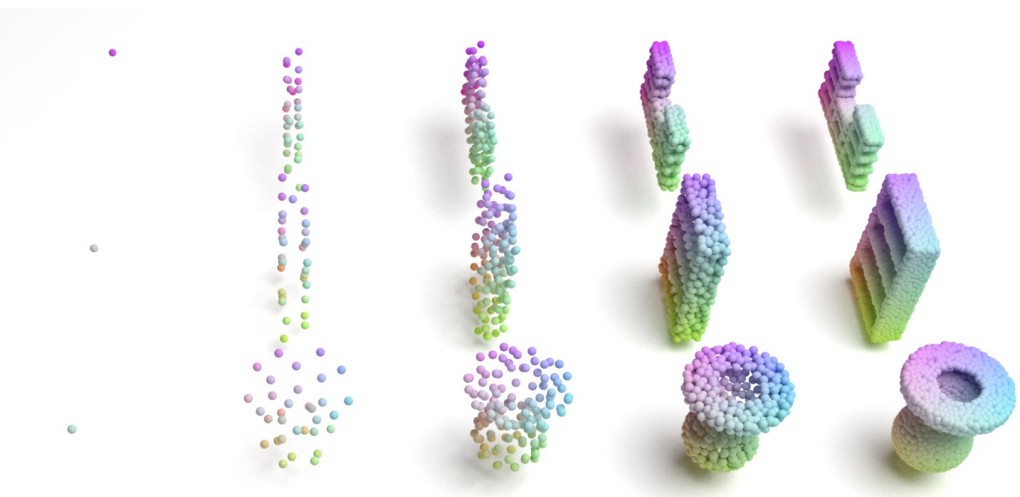

Figure S9: Illustration of multi-scale point cloud generation from the *PointNSP* model.

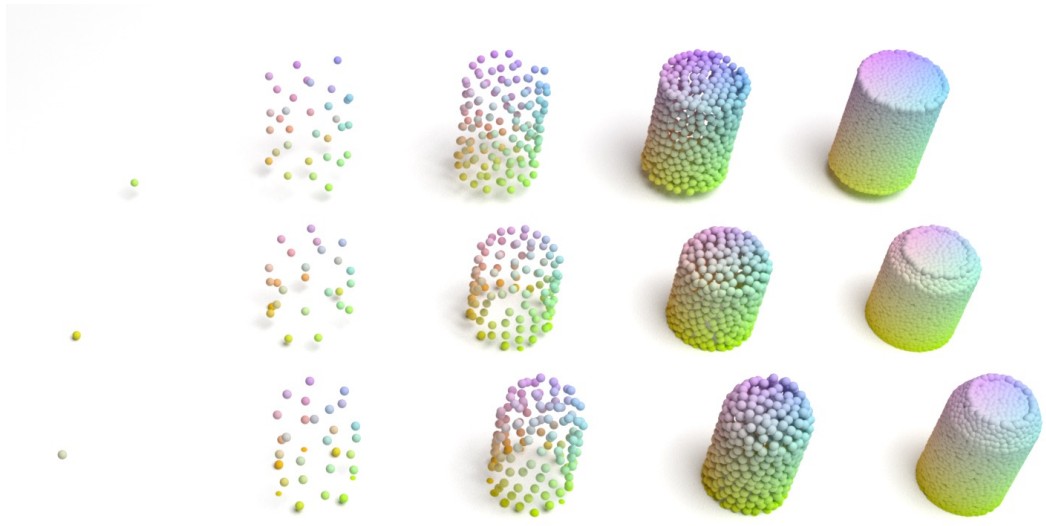

Figure S10: Illustration of multi-scale point cloud generation from the *PointNSP* model.

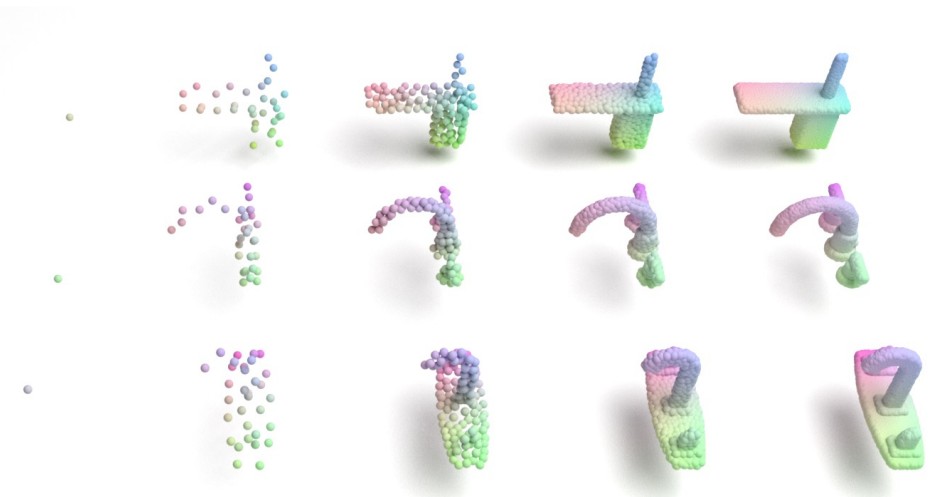

Figure S11: Illustration of multi-scale point cloud generation from the *PointNSP* model.

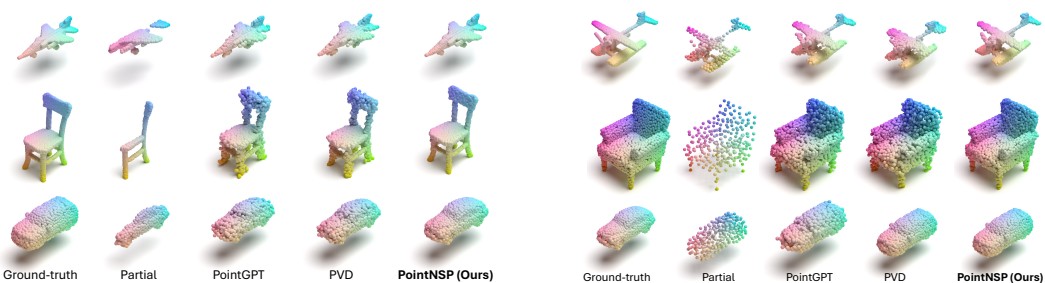

Figure S12: (Left) Visualizations of point cloud completion results. (Right) Visualizations of point cloud upsampling results.

