# OpenReview forum: "PoinNSP: Autoregressive 3D Point Cloud Generation with Next-Scale Leve-of-Detail Prediction"
_ICLR.cc/2026/Conference — ICLR 2026 Conference Withdrawn Submission_

### Official Review · Reviewer_PmGo · 2025-10-26

**Soundness:** 3
**Presentation:** 3
**Contribution:** 2
**Rating:** 6
**Confidence:** 3

**Summary:**

The paper proposes PointNSP, an autoregressive model for 3D point cloud generation that predicts shapes via next-scale level-of-detail (LoD) refinement instead of point-by-point orderings. This design preserves permutation invariance and improves efficiency. On ShapeNet, PointNSP achieves state-of-the-art quality and outperforms diffusion-based baselines in speed and scalability.

**Strengths:**

1. The next-scale generation idea is innovative and effectively ensures permutation invariance, addressing a key limitation of next-point autoregressive models.
2. The authors demonstrate strong scalability by testing the model on higher-resolution point clouds, clearly showing that the multi-scale approach remains effective as point density increases.
3. The experiments are comprehensive, and the ablation studies are well-structured, providing clear evidence for the contribution of each component.

**Weaknesses:**

1. What is the overhead for the decoder? Each step decodes the point cloud to get the positional encoding, which seems not optimal. Why doesn’t the codebook contain the positional encoding information itself, instead of needing to decode into absolute points?
2. How many steps are used for the diffusion-based generation method? What solver is used in that paper? Advanced solvers could be 10× faster. If an autoregressive model needs 10 steps and the advanced solver also needs 10 denoising steps, does that mean the runtime is similar?
3. When the scale equals 1, how does this single point’s position affect the generation results?
4. What is the setup for the point cloud completion task in PointNSP? Does it treat the partial point cloud at some centrain scale and generate full point clouds based on that scale?

**Questions:**

See weakness

---

### Official Review · Reviewer_M2AB · 2025-10-30

**Soundness:** 2
**Presentation:** 2
**Contribution:** 2
**Rating:** 4
**Confidence:** 5

**Summary:**

This paper proposes an autoregressive generative model for 3D point cloud synthesis, eliminating the need for an explicit point ordering. The method progressively refines point clouds from coarse to fine granularity, enabling high-quality 3D shape generation. A two-stage training paradigm is employed: first, the model learns multi-scale discrete representations of point clouds; then, an autoregressive transformer is trained to predict the next finer scale based on the learned hierarchy.

**Strengths:**

（1）The work successfully migrates the next-scale prediction paradigm sourced from image generation to the domain of 3D point cloud generation.
（2）The proposed framework has superior computational efficiency to conventional point-wise autoregressive approaches, which enhances its practicality.

**Weaknesses:**

（1）One of my primary concerns pertains to the novelty of the work. The core idea mainly involves a direct adaptation of the VAR framework to 3D point clouds, with relatively limited technical innovations.
（2）The authors categorically assert that autoregressive point cloud generation has long lagged behind diffusion-based
 approaches in quality, attributing this gap to the fact that autoregressive models impose an artificial order on unordered point sets—yet they fail to provide clear evidence to support this claim.
（3）Compared to other methods, the performance improvement of this paper on the ShapeNet dataset seems rather limited. The study uses only a single dataset and considers only three object categories.

**Questions:**

As shown in Table 4, why does the conditional generation experiment only include three categories (airplane, chair, car)? I cannot help but wonder whether the method can still achieve optimal performance in scenarios involving other categories.

---

### Official Review · Reviewer_WpLB · 2025-11-01

**Soundness:** 3
**Presentation:** 3
**Contribution:** 2
**Rating:** 6
**Confidence:** 4

**Summary:**

This paper succsessfully adopt the next-scale autoregressive prediction paradigm previously used in image generation to point generation. The results show the paradigm can generate bigger number of cloud points as well as better quality point cloud. Extensive comparisons and ablation studies have been shown to demonstrate the effectiveness of the work.

**Strengths:**

1. Framework design is intuitive and easy to follow: Next scale autoregressive prediction has been validated in the 2D image geneneration task. Adopting this to point cloud generation is natural.

2. This work has made substatial effort to validate design choices in the ablation study section. Namely Table 3.

3. Extensive comparisons have been made in the Table 1. This sufficiently shows the effectiveness of proposed new point cloud generation framework.

**Weaknesses:**

1. **Presentation of the paper**:

     a. On point cloud generation, is there a topology thing? Mentioning **consistent topology** in the abstract sounds confusing to me.
&nbsp;

2. "Concern on the memorization issue":
     We do see higher quality of generating shapes, especially we see more smooth surface may be recovered from the generated points cloud. However, since ShapeNet is a small dataset, it is recommended to find a nearest neighbour within the training set for each of the generated samples.

&nbsp;

3. "Concern on the actual application":
     This work is trying to generate point clouds in a higher resolution. The point clouds genearted now are pure positions. The ultimate product, I am supposing, will be surfaces of the object extracted from the point clouds (via some other algorithms). Based on this thought line, it is better show how the extracted surfaces look like. It will be treasured if it can compare the surface quality with Shape2VectorSet [1], so that we can get a sense what kinds of geometry details we can get out of it.

[1] Zhang, Biao, et al. "3dshape2vecset: A 3d shape representation for neural fields and generative diffusion models." ACM Transactions On Graphics (TOG) 42.4 (2023): 1-16.

**Questions:**

1. As the weaknesses section points out:

    a. Can you show the nearest neighbour search on generated samples within the training dataset?

    b. Could you please do the generation experiment on Objaverse to see if your framework can be extended to a much larger dataset?

    c. Please show the extracted surfaces from the generated point clouds.

    d. The point cloud completion task can also be compared with method such of Shape2VectorSet[1] line of works.


[1] Zhang, Biao, et al. "3dshape2vecset: A 3d shape representation for neural fields and generative diffusion models." ACM Transactions On Graphics (TOG) 42.4 (2023): 1-16.

---

### Official Review · Reviewer_T61j · 2025-11-01

**Soundness:** 2
**Presentation:** 1
**Contribution:** 2
**Rating:** 2
**Confidence:** 3

**Summary:**

The paper presents a novel approach to point clouds generation. Inspired by recent multi-scale next token prediction works in images, this paper suggest a multi-scale autoregressive generation for points clouds as well. Formulate the tokenization, and novel architecture, and present state-of-the-art results on standard point clouds generation benchmarks.

**Strengths:**

(+) Novel application, bring the concept of multi-scale next token prediction to 3d point clouds generation. With all the subtleties required for this transition to work.

(+) State-of-the-art results

**Weaknesses:**

the work is generally poorly written and difficult to understand

It was very hard to understand the tokenization process, and the generation process. I didnt understand it yet. I do believe it is valid, thus requesting for major clarifications.
* Figure 3 - what do each arrow mean? Add clear explanation to every detail in the figure, in its caption.
* Line “201” “where FPSk denotes a nested operation with k loop” without knowing well previous works and deducing a possible explanation of this sentence, impossible to understand.
* Line 230: Whats f^i? Is that the corresponding f feature of the sampled i point from the s_k sampled points in k phase? If so, mention that explicitly. Whats f_k? Is that a vector of all of the encoded features for the s_k points sampled for k? If so, mention that explicitly, and its dimensions.
* Line 215 - whats [V]?
* Equation 4 - if my previous assumptions regarding f dimensionality were correct, this equation doesnt make sense. How can f_0 of dimension s_0 = number of points (am I even right here?) be an addition of f_1 vector which is of dimension s_1, when s1<s_K?
* Equation 4 - How can f=f_0, when the highest resolution is s_K and not s_0?
* Equations 4 and 5 - what is the upsampling process? Does it add points to predict the same number of points as in the original X? I didnt understand. How does it work if so? Add visualization of these process.
* How exactly do you use the upsampeld points of a specific step for the next step? I didnt understand
* Figure 4 - How exactly are the input processed into the transformat? You show there an arrow inserting plain X points into the transformer next input. How? Why aren’t them upsampled as you said earlier? Didnt understand that. Explain clearly, both in text and in the figure caption.
* Equation 8, The position aware softmax - sounds interesting. But how does it work?  Whats W_p? Is it learned? Is P_k of the dimensions you wrote there s_k x d? If so, it means that P_k = f_k, isn’t it? If so, how can that be positional encoding? F doesnt have the position in it does it? Maybe it is another d, of encoding of sinusoidal position encoding as in line 316, and the d is another d, of explicitly stated positional encoding function? I read the appendix, and thus guessed it, but still not sure as the notations doesnt seem like I am right. If my guess here is correct, it cannot be understood from the article
* Algorithm 1, - add caption. The operations there cannot be understood until you read the entire method section, and it is presented first.  i'm worried most readers will read it and not understand anything, and later understand he missed what are the query, upsampling, encoding etc operations there

**minor**

* Line 196 “Importantly, we do not fix the initial starting point of FPS, so the sampled points in Xk vary during training.” - what does it mean? I guess it means that in every new epoch, the fps is with a different seed. If so, hard to understand.

* Line 51 “their Markovian assumption overlooks global context, often leading to incoherent shapes” Why diffusion models overlook global context? I am pretty sure that is wrong, as they gradually refine the entire point cloud, they are always aware of the entire context.

* Line 47: “permuting the points leaves the shape unchanged”. In what way such permutation leaves them unchanged, what kind of permutation. Permutation in the order they are inserted to the model? They are un-ordered in the first place. A bit confusing phrasing.

* Missing some basic ablations. Can enhance the paper contribution. E.g. the position aware softmax affect on performance.

**Questions:**

see weaknesses

**Details Of Ethics Concerns:**

a very similar work was found on arxiv: https://arxiv.org/pdf/2503.08594
I haven’t explored it in depth, but it appears that many of its ideas are repeated in the current submission. This prior work is not mentioned in the paper. Given the strong resemblance, I would expect at least a citation and a clear discussion of the differences.
I wasn’t entirely sure how to address this, and I also did not find evidence that the previous work has been published elsewhere. Still, I think we should look into it more carefully.

---

### Note · Authors · 2025-11-14

**Comment:**

We would like to express our sincere gratitude to the reviewers for their thoughtful and constructive feedback. We genuinely appreciate the effort invested in evaluating our work and had hoped to further interact with the reviewers to improve the manuscript based on their insights. However, due to certain constraints, we regret to withdraw the submission at this time. We will carefully incorporate the reviewers’ comments to strengthen the work moving forward.

**Withdrawal Confirmation:**

I have read and agree with the venue's withdrawal policy on behalf of myself and my co-authors.